# MHC-compatible bone marrow stromal/stem cells trigger fibrosis by activating host T cells in a scleroderma mouse model

Yoko Ogawa[1†], Satoru Morikawa[2,3†], Hideyuki Okano[3], Yo Mabuchi[3,4], Sadafumi Suzuki[3], Tomonori Yaguchi[5], Yukio Sato[3,6], Shin Mukai[1], Saori Yaguchi[1], Takaaki Inaba[1], Shinichiro Okamoto[7], Yutaka Kawakami[5], Kazuo Tsubota[1], Yumi Matsuzaki[8*], Shigeto Shimmura[1*]

[1]Department of Ophthalmology, Keio University School of Medicine, Tokyo, Japan; [2]Department of Dentistry and Oral Surgery, Keio University School of Medicine, Tokyo, Japan; [3]Department of Physiology, Keio University School of Medicine, Tokyo, Japan; [4]Department of Biochemistry and Biophysics, Graduate School of Health Care Sciences, Tokyo Medical and Dental University, Tokyo, Japan; [5]Division of Cellular Signaling, Institute for Advanced Medical Research, Keio University School of Medicine, Tokyo, Japan; [6]Department of Emergency and Critical Care Medicine, Keio University School of Medicine, Tokyo, Japan; [7]Division of Hematology, Department of Internal Medicine, Keio University School of Medicine, Tokyo, Japan; [8]Department of Life Science Laboratory of Tumor Biology, Faculty of Medicine, Shimane University, Izumo, Japan

*For correspondence: matsuzak@
med.shimane-u.ac.jp (YM); shige@
z8.keio.jp (SS)

[†]These authors contributed
equally to this work

Competing interest: See
page 17

Reviewing editor: Ronald N
Germain, National Institute of
Allergy and Infectious Diseases,
United States

**Abstract** Fibrosis of organs is observed in systemic autoimmune disease. Using a scleroderma mouse, we show that transplantation of MHC compatible, minor antigen mismatched bone marrow stromal/stem cells (BMSCs) play a role in the pathogenesis of fibrosis. Removal of donor BMSCs rescued mice from disease. Freshly isolated PDGFRα[+] Sca-1[+] BMSCs expressed MHC class II following transplantation and activated host T cells. A decrease in FOXP3[+] CD25[+] Treg population was observed. T cells proliferated and secreted IL-6 when stimulated with mismatched BMSCs in vitro. Donor T cells were not involved in fibrosis because transplanting T cell-deficient RAG2 knock out mice bone marrow still caused disease. Once initially triggered by mismatched BMSCs, the autoimmune phenotype was not donor BMSC dependent as the phenotype was observed after effector T cells were adoptively transferred into naïve syngeneic mice. Our data suggest that minor antigen mismatched BMSCs trigger systemic fibrosis in this autoimmune scleroderma model.

## Introduction

Systemic fibrosis is a feature of autoimmune disease such as systemic sclerosis (SSc) or Sjögren's syndrome involving exocrine glands (*Ferrara et al., 2009*; *Filipovich et al., 2005*). A mouse model for human SSc reported by Zhang et. al. involves transplantation of B10.D2 bone marrow into MHC matched, minor antigen mismatched BALB/c host (*Zhang et al., 2002*). This model of SSc occurs spontaneously without the use of artificial agents such as bleomycin (*Yamamoto and Nishioka, 2004*), and exhibits characteristics of human SSc including fibrosis, inflammation, and autoimmunity. Animal models are effective in screening for therapeutic interventions such as anti IL-6 (*Le Huu et al., 2012*) and angiotensin II type-1 receptor antagonists (*Yaguchi et al., 2013*). However, such a

**eLife digest** Systemic scleroderma is an autoimmune disease caused by the immune system attacking the body's connective tissues, which provide the body with structural support. Immune cells called T cells accumulate in connective tissue, which leads to the hardening of the skin and may also damage the heart, lungs and other internal organs. However, it is not clear what prompts the T cells to accumulate in the connective tissues of these individuals.

Autoimmune diseases develop when the immune system mistakenly identifies host cells as being a threat to the body. Normally, the immune system recognizes healthy body cells by the presence of particular proteins on the surface of the cells. A set of surface proteins called the major histocompatibility complexes (MHCs) play a major role in this process, but there are also many other surface proteins that play more minor roles.

In 2002, researchers developed a method that can trigger the symptoms of systemic scleroderma in mice. This method involves transplanting bone marrow from one mouse into another mouse. Both mice have identical MHC proteins on the surfaces of their cells, but have some differences in other cell surface proteins, and so the bone marrow from the donor mouse triggers an immune response in the recipient.

To better understand how this mouse "model" of systemic scleroderma works, Ogawa, Morikawa et al. refined the method so that they could just transplant specific types of bone marrow cells into the recipient mice. The experiments reveal that bone marrow stromal stem cells, but not so-called "hematopoietic stem cells", from a donor mouse are responsible for triggering the immune response and disease symptoms in the recipients.

Ogawa, Morikawa et al.'s findings show that mismatched minor cell surface proteins on bone marrow stromal stem cells can trigger symptoms of systemic scleroderma in mice. Further studies are required to find out how these cells encourage T cells to trigger an autoimmune response.

spontaneous model is also a valuable tool for investigating the pathogenesis of SSc, which is still largely unknown.

In order to shed light onto the mechanisms leading to fibrosis in this SSc mouse model, it is necessary to isolate the different cellular fractions within the B10.D2 donor bone marrow, namely, hematopoietic stem cells (HSCs) and bone marrow stromal/stem cells (BMSCs). Multipotent BMSCs in the bone marrow differentiate into several mesenchymal lineages including fibroblasts, adipocytes, osteocytes, and chondrocytes (*Pittenger et al., 1999*; *Prockop, 1997*). However, due to the lack of specific markers, a crucial step involving in vitro expansion was required to isolate BMSCs, which may modify their phenotype and function (*Banfi et al., 2000*). Most current information on BMSCs comes from such in vitro studies of adherent cells referred to as fibroblast CFUs (CFU-Fs) (*Conget and Minguell, 1999*; *Friedenstein et al., 1974*; *Pittenger et al., 1999*; *Prockop, 1997*), which are a heterogeneous population of cells at best. Therefore, the in vivo dynamics of BMSCs after whole bone marrow transplantation (WBMT) are still unknown, and the establishment of a solid experimental system to trace the fate of BMSCs following transplantation was required.

In order to establish an animal model with traceable donor BMSCs and HSCs, we applied our previously reported method for prospectively isolating murine BMSCs based on their expression of PDGF receptor $\alpha$ and Sca-1 (PDGFR$\alpha^+$/ Sca-1$^+$ (P$\alpha$S) cells) (*Morikawa et al., 2009a*). Selectively isolated P$\alpha$S-BMSCs without in vitro expansion represents highly clonogenic and multi-potent population of cells including hematopoietic niche cells, osteoblasts, and adipocytes after systemic in vivo transplantation (*Morikawa et al., 2009a*; *2009b*). Our model allows for the first time both whole bone marrow transplantation, as well as the selective transplantation of freshly isolated BMSCs and/or HSCs into recipient mice. By applying this modified SSc model using prospectively isolated BMSCs and HSCs, we sought to identify the role of donor HSCs and BMSCs in the pathogenesis of the autoimmune-related fibrosis in SSc. Here, we show how mismatched donor BMSCs not only contribute to fibrosis in various organs, but also trigger the onset of autoimmune disease by activating host T cells.

# Results

## Mismatched PαS-BMSCs is a key inducer of fibrosis after bone marrow transplantation

A known bone marrow transplantation (BMT) model using 8-week-old donor B10.D2 (H-2d) mice and recipient BALB/c mice (H-2d), is a MHC-compatible, minor histocompatibility antigen (miHA)–incompatible model of systemic fibrosis (*Figure 1—figure supplement 1A*) (*Zhang et al., 2002*). Signs of fibrosis appear by 3 weeks after BMT and progresses to a full-blown disease by 8 weeks characterized by excessive fibrosis of the lacrimal gland, conjunctiva, salivary gland, skin, lung, liver, and intestine (*Figure 1—figure supplement 1B,C*).

In order to elucidate the source and role of donor-derived fibroblasts, we first modified the SSc model by co-transplanting prospectively isolated HSCs with prospectively isolated BMSCs. PαS-BMSCs and side population (SP-HSCs) from B10.D2 or BALB/c were individually isolated (*Figure 1A* and *Figure 1—figure supplement 2*). We confirmed that PαS BMSCs cells expressed CD29, CD90, and CD106, known as established markers of BMSCs. However, CD31 and CD133, markers of endothelial cells, consisted of only a small portion of the PαS BMSCs fraction (*Figure 1—figure supplement 2C*). These PαS-BMSCs and side population (SP-HSCs) from B10.D2 or BALB/c were systemically co-transplanted into BALB/c recipients as defined combinations (*Figure 1Bi–iv*). Unlike the original model using WBM, the graft does not include donor type mature hematopoietic and non-hematopoietic cells, and the cells were provided from each lineage of stem cell after bone marrow engraftment.

Notably, progressive fibrosis was only observed in mice receiving mismatched BMSCs (*Figure 1Bii,iv* and *Figure 1—figure supplement 3Aii,iv*). The recipients of mismatched HSCs combined with syngeneic BMSCs were indistinguishable from syngeneic HSC and syngeneic BMSC recipients. Physicological connective tissues are also stained in blue in the lacrimal gland, salivary gland, and intestine when BALB/c mice received mismatched HSCs and syngeneic BMSCs (*Figure 1Bi,iii* and *Figure 1—figure supplement 3Ai,iii*). Since heat-shock protein 47 (HSP47), a molecular chaperon specific to collagen-secreting cells, has been reported as a useful marker for fibroblasts in paraffin-embedded tissue sections (*Kuroda and Tajima, 2004*), we determined the state of fibrosis by (1) semi-quantitative analysis to count the number of HSP47+ fibroblasts per field and (2) Mallory staining, a specific marker of connective tissue (*Figure 1B,C* and *Figure 1—figure supplement 3A, B*) (*Brack et al., 2007*). Fibrotic lesions in mismatched BMSC recipients had significantly higher HSP47[+] fibroblasts (*Figure 1C*). Lacrimal gland function was also reduced only in mismatched BMSC recipients (*Figure 1D*). In addition, all these observations were canceled in recipients of BMSC-depleted WBM transplantation (*Figure 1Bv,Cv,Dv* and *Figure 1—figure supplement 3Av,Bv*). Minimal Mallory staining areas are physiological changes and necessary to support the structure of ducts and intestinal walls (*Figure 1—figure supplement 4*). The severity of fibrosis triggered by mismatched BMSCs was dose dependent on the number of transplanted BMSCs (*Figure 1E,F*). These results suggest that mismatched donor BMSCs or their progeny triggered the onset of disease.

Cells double positive for HSP47 and enhanced green fluorescent protein (EGFP) were detected in fibrotic tissue when mismatched EGFP[+]B10.D2 PαS-BMSCs were transplanted with wild-type B10. D2 SP cells (*Figure 2A*). Approximately $42.9 \pm 7.3\%$ of the HSP47[+] cells in fibrotic tissue were GFP[+], indicating that the majority of fibroblasts were derived from donor PαS-BMSCs (*Figure 2A*). On the other hand, a notable number of cells derived from mismatched donor EGF[+]SP-HSCs also migrated to fibrotic lesions, but none of the GFP[+] cells expressed HSP47 (*Figure 2B*). The lineage exclusivity of this transplantation excludes the possibility that HSP47[+] fibroblasts were derived from SP-HSCs as shown in our previous report (*Koide et al., 2007*). The accumulation of donor-derived fibroblasts was not observed in syngeneic transplantation, where fibrosis does not occur (*Figure 2C*). HSP47 expression is clearly seen especially in activated fibroblasts (*Figure 2A*), but very faint expression is detected in quiescent fibroblasts in the syngeneic BMSCs transplanted recipients' target organs (*Figure 2C*). These results suggest that mismatched BMSCs migrate into the target organs and proliferate under pathological microenvironment.

In vitro culture also confirmed that the majority of fibroblasts from lacrimal gland cultures were derived from mismatched EGFP[+] BMSC (*Figure 2D*, right), whereas no EGFP[+] fibroblasts were observed after mismatched EGFP[+] HSC transplantation (*Figure 2D*, left). This suggests that the

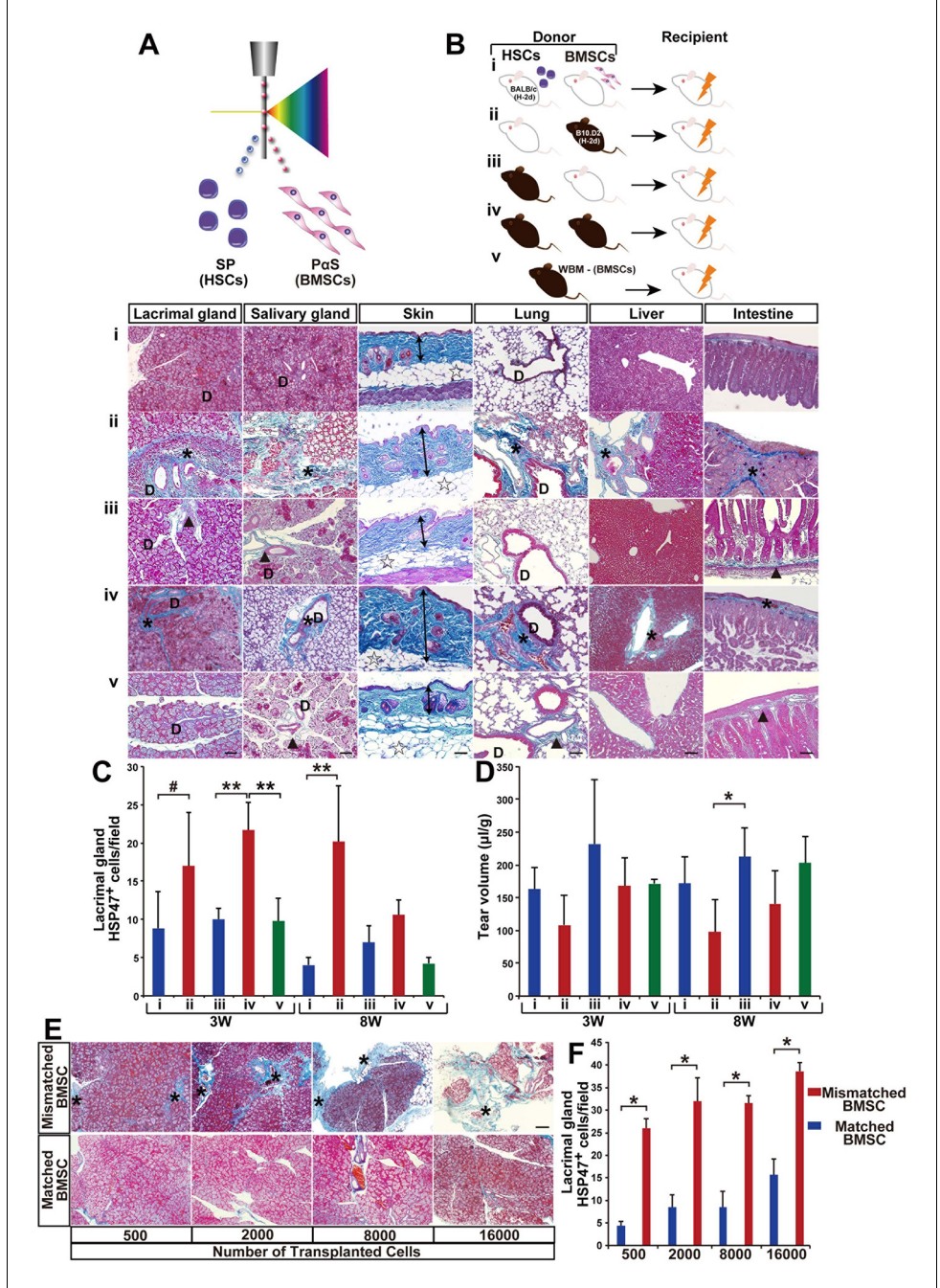

**Figure 1.** Modified SSc model by co-transplanting isolated HSCs and BMSCs. (**A**, **B**) Transplantation model with different combinations of syngeneic (white) and mismatched (black) HSCs and BMSCs co-transplanted into BALB/c mice. (**B**, i–v) (i) Negative control, (ii) mismatched BMSC transplantation, (iii) syngeneic BMSC transplantation, (iv) positive control, and (v) mismatched MSC-depleted WBMT. Excessive fibrosis (deep blue, and *) in various organs was observed in mismatched BMSC transplanted mice after 3 weeks. Double arrows indicate epidermal and dermal thickness. The fibrotic areas were assessed as the ratio of the blue-stained area per field. D, duct. Minimal Mallory staining areas (▲) in (iii) and (v) are physiological changes and necessary to support the structure of ducts and intestinal walls. Scale bar, 100 μm (50 μm in liver). (**C**) HSP47+ fibroblasts in the lacrimal glands were significantly higher following mismatched BMSC transplantation (red) compared to syngeneic BMSC (blue), and BMSC-depleted WBM transplantation (green). The number of HSP47+ cells in negative control (i) (blue), and syngeneic MSC transplantation (iii) (blue), and mismatched MSC transplantation (ii) (red), and positive control (iv) (red), and mismatched MSC-depleted WBMT (green). Data are shown as mean ± SD, #p<0.05, *p<0.01, **p<0.001. (**D**) Tear volume in the same groups described in (**C**). Data are shown as mean ± SD, n = 2-5 per group, *p<0.05. (**E**, **F**) The degree of fibrosis (blue) in the lacrimal glands of mismatched BMSC recipients was dose dependent. Excessive fibrotic areas are shown in deep blue (*). Data are shown as mean ± SD, n = 3. Scale bar, 100 μm. *p<0.001. BMSCs, bone marrow stromal/stem cells; HSCs, hematopoietic stem cells; SD, standard deviation.

*Figure 1 continued on next page*

*Figure 1 continued*

The following source data and figure supplements are available for figure 1:

**Source data 1.** HSP47[+] cells/ field in the lacrimal glands.
**Source data 2.** Tear volume following transplantation.
**Source data 3.** The number of HSP47[+] cells/field in the lacrimal glands after transplantation of 500, 2000, 8000, and 16,000 syngeneic or mismatched BMSCs shown in (F).
**Source data 4.** Number of HSP47[+] cells/ field from in various target organs after whole bone marrow transplantation.
**Source data 5.** HSP47[+] cells per field in the salivary gland, skin, lung, liver, and intestine shown in *Figure 1—figure supplement 3B*.
**Figure supplement 1.** Autoimmune-associated fibrosis following whole bone marrow transplantation in mouse model.
**Figure supplement 2.** Flow cytometry protocol for Isolating BMSCs and HSCs.
**Figure supplement 3.** Modified SSc model by co-transplanting isolated HSCs and BMSCs in other target organs.
**Figure supplement 4.** Mallory staining of normal organs.

migration of donor-derived fibroblasts to peripheral organs is one of the characteristic phenomena associated with pathogenesis or progression of this disease.

## Host T cells are required for the progression of fibrosis

The progression of fibrosis in this mouse model is T-cell-dependent as evidenced by the fact that fibrosis was not observed when BALB/c-nu/nu mice were used as recipients of mismatched B10.D2 BMSCs transplantation (*Figure 2—figure supplement 1*). The next question was whether these pathogenic T cells were donor-derived T cells, or residual host T cells that lost self-tolerance. Both host-derived T cells and donor HSC-derived T cells are believed to exist in this mouse model, because mature T cells often escape radiation damage and remain in the peripheral blood of recipients (*Anderson et al., 2004*). Interestingly, almost all spleen cells were donor-derived in matched WBMT, whereas the remaining host-derived cells after mismatched WBMT were higher in mismatched WBMT (*Figure 2E*).

To determine the origin of T cells, we used BALB/c-RAG2KO mice as recipients or donors of HSCs. In the former lacking recipient T cells (*Figure 3A*), the fibrosis was not observed (*Figure 3A, C*). However, donor RAG2KO HSCs lacking T cells combined with mismatched BMSCs in the latter (*Figure 3B*) induced fibrosis with increased HSP47[+] fibroblasts in wild-type BALB/c recipients (*Figure 3B,C*). In addition, we found that recipient-derived Th 17 cells were in proximity of donor-derived BMSCs (*Figure 4A*, left) and that donor BMSCs produce IL-6 (*Figure 4B*, left) around the ducts of the lacrimal gland in BALB/c recipient mice at 8 weeks after mismatched BMSC transplantation. In comparison, mismatched donor HSC did not produce Th 17 cells (*Figure 4A*, right) or IL-6 (*Figure 4B*, right).

In spleen cells, flow cytometric analysis revealed a significant increase of recipient-derived CD4[+]Th 17 cells when donor BMSCs with RAG2KO HSCs were transplanted into BALB/c recipients (*Figure 4C*, Left). In contrast, CD4[+]Th 17 cells were scarcely detected when donor BMSCs with wild-type BALB/c HSCs were transplanted into RAG2KO recipient (*Figure 4C*, Right).

To further confirm the function of mismatched PαS-BMSC-transplanted recipient-derived T cells, we adoptively transferred splenic T cells from mismatched BMSC-transplanted recipient into naive nude mice (BALB/c background). We found the primary inflammation and following fibrosis occurred in the target organs including the lacrimal glands, salivary glands, skin, lung, liver, and intestine (*Figure 5A*, *Figure 5—figure supplement 1*). The number of HSP47[+] fibroblasts per field was significantly increased in the adoptively transferred nude mice (*Figure 5B*) compared with the wild-type control. Splenic PBMC of the adoptive transferred recipients revealed that CD4+Th 17 cells were markedly elevated (*Figure 5C*, Left) compared to the wild-type BALB/c background nude mice

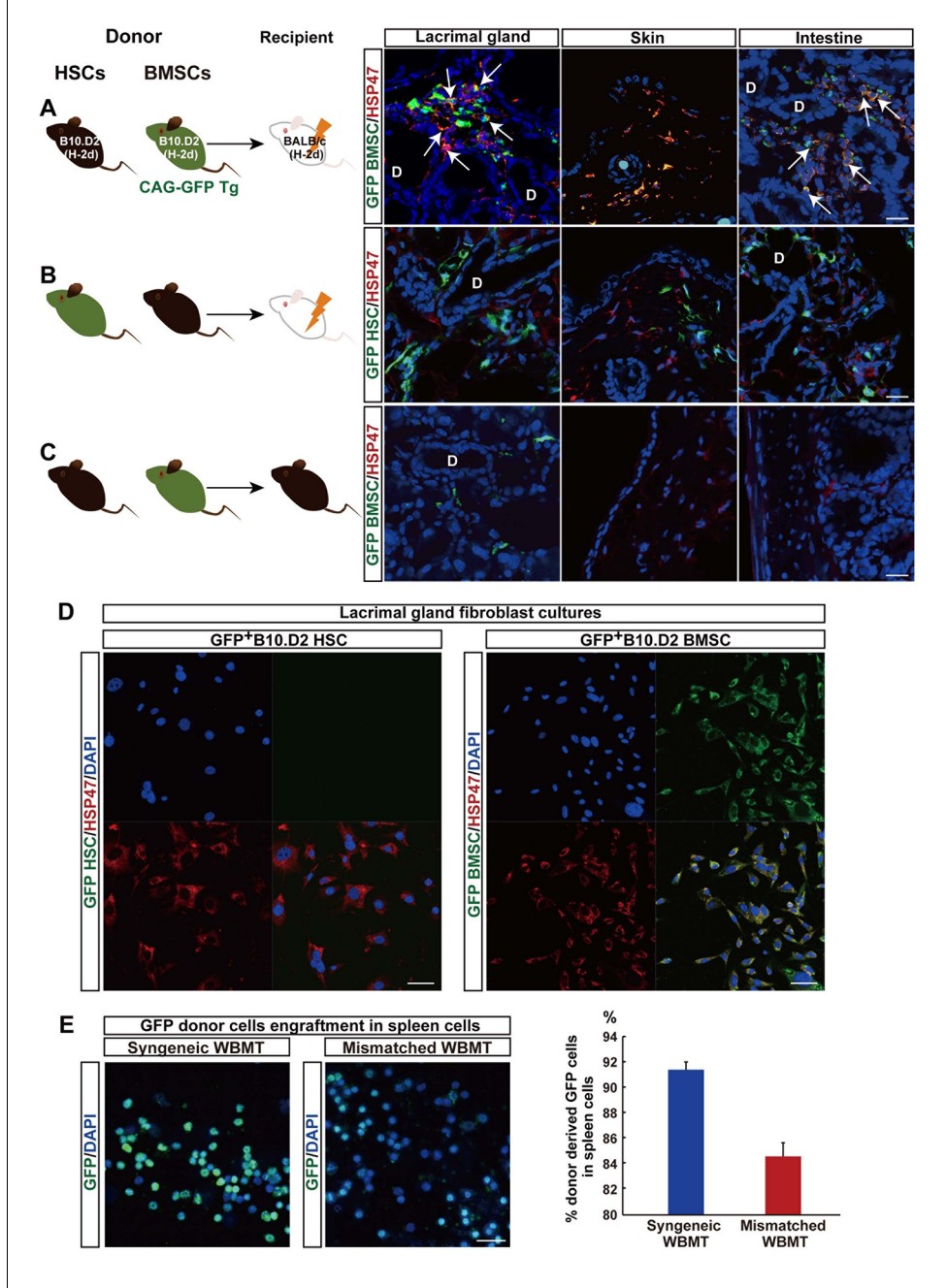

**Figure 2.** Mismatched donor derived-fibroblasts engrafted in target tissues of fibrosis. (A, B) Transplantation scheme of EGFP[+]-labeled B10.D2 PαS-BMSCs (A) or SP-HSCs (B). Each was co-transplanted into wild-type BALB/c mice along with unlabeled SP-HSCs (A) or PαS-BMSCs (B), respectively. Arrows indicate colocalized cells in yellow (GFP-labeled BMSCs expressed HSP47) in the lacrimal gland and intestine. (C) Syngeneic wild-type B10.D2 SP-HSCs and EGFP[+]-labeled B10.D2 PαS-BMSC co-transplantation into wild B10.D2 mice. HSP47[+] (red) fibroblasts observed in mismatched BALB/c recipients were BMSC-derived, and not HSC cells. Nuclei were counter stained with DAPI (blue). The data in *Figure 2A–C* from two replicate experiments (n = 3 per group). Scale bar, 20 μm. (A–C) D, duct. (D) Cultured HSP47[+] lacrimal gland fibroblasts after mismatched EGFP[+] HSC transplantation were EGFP[-] (left), while the majority of fibroblasts after mismatched EGFP[+] BMSC transplantation were EGFP[+] fibroblasts (right, positive cells in yellow) indicating the donor BMSC origin of fibroblasts. (E) Donor–derived EGFP[+] cells were observed in the spleen 3 weeks after syngeneic EGFP[+] WBMT (left), while engraftment of donor cells were sparse following mismatched WBMT (right), indicating a number of residual host cells remained after mismatched WBMT. *Figure 2D,E* from representative data of two replicate experiments (n = 2 or 3 per group). Scale bar, D = 50 μm, E = 20 μm. BMSC, bone marrow stromal/stem cells; HSP, heat-shock protein; HSCs, hematopoietic stem cells

The following source data and figure supplement are available for figure 2:

*Figure 2 continued on next page*

*Figure 2 continued*

**Source data 1.** Percentage of donor–derived EGFP+ cells in the spleen 3 weeks after EGFP+ WBMT.
**Figure supplement 1.** Mismatched PαS-BMSCs do not induce sclerodermatous fibrosis in recipient nude mice.

(*Figure 5C*, Right). We conducted IL-17 ELISA using the serum samples from the same mice of the experiments shown in *Figure 4C*. IL-17 concentration significantly increased in the serum from the adoptive transferred recipients group compared with the control group (*Figure 5D*).

These results strongly suggest that recipient mature T cells act as an effector for mismatched PαS-BMSCs during the onset of fibrosis. Since EGFP+ BMSC-derived cells were not observed in the cortical region of the thymus within this time frame (data not shown), it is unlikely that escape of donor-derived T cells from negative selection by the host thymus is involved in the recognition of host tissue as non-self.

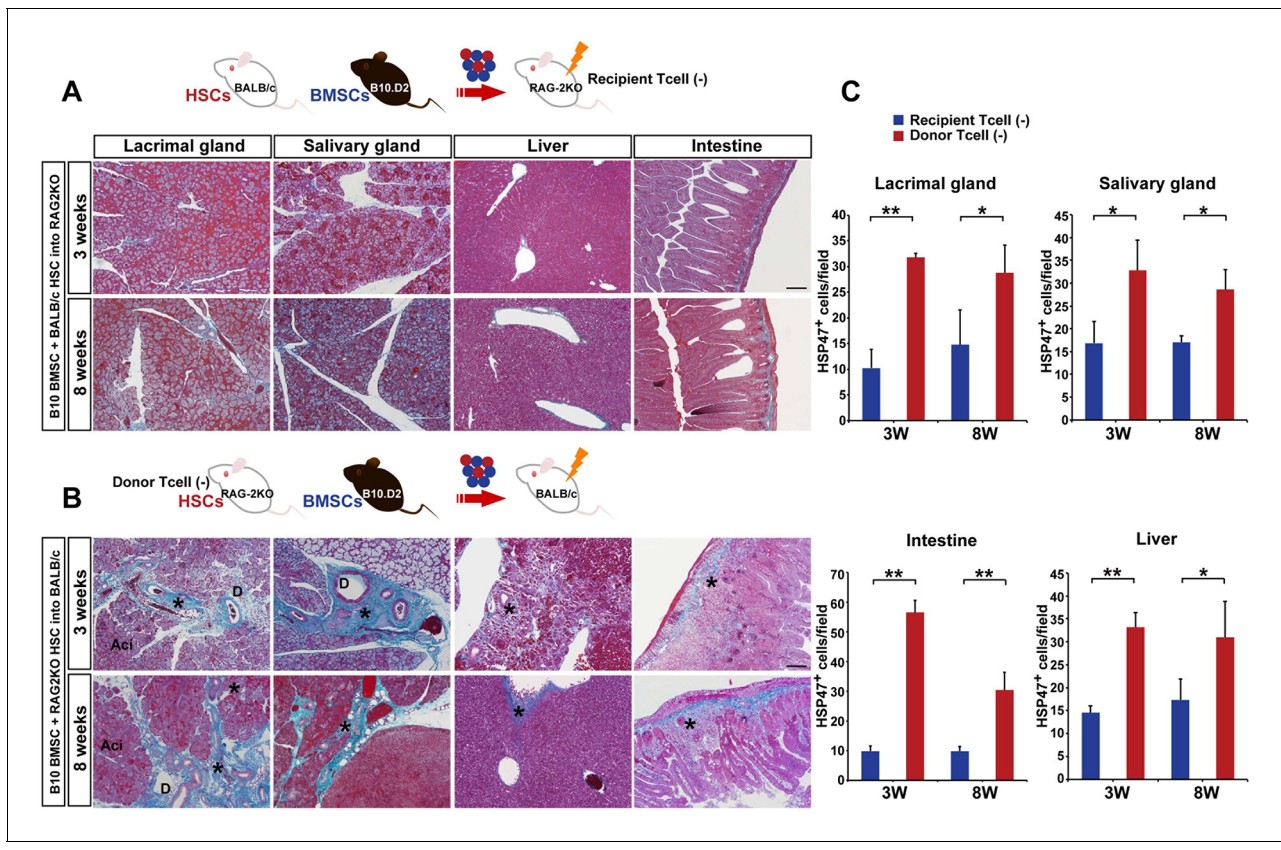

**Figure 3.** Host T cells are required for the progression of fibrosis. (**A**) Mallory staining of recipient RAG2KO organs shows that transplantation of mismatched BMSCs did not induce fibrosis in the absence of recipient T cells. Scale bar, 100 μm. (**B**) Tissue inflammation and excessive fibrosis in deep blue (*) was observed in BALB/c recipient mice after B10.D2 BMSC and RAG2KO HSC transplantation, despite the lack of donor T cells. Data collected from two replicate experiments (n = 3 per group). Scale bar, 100 μm. (**A**, **B**) D, duct; Ac, Acinus. (**C**): Significantly higher number of HSP47+ fibroblasts was observed in BALB/c recipients after B10 BMSC + RAG2KO HSC transplantation (red) compared to RAG2KO recipients after B10 BMSC + BALB/c HSC transplantation (blue). Data from five different fields in two replicate experiments. Data are shown as mean ± SD. *p<0.01, **p<0.001.
The following source data is available for figure 3:

**Source data-1.** Number of HSP47+ cells per field from the lacrimal gland, salivary gland, liver, and intestine.

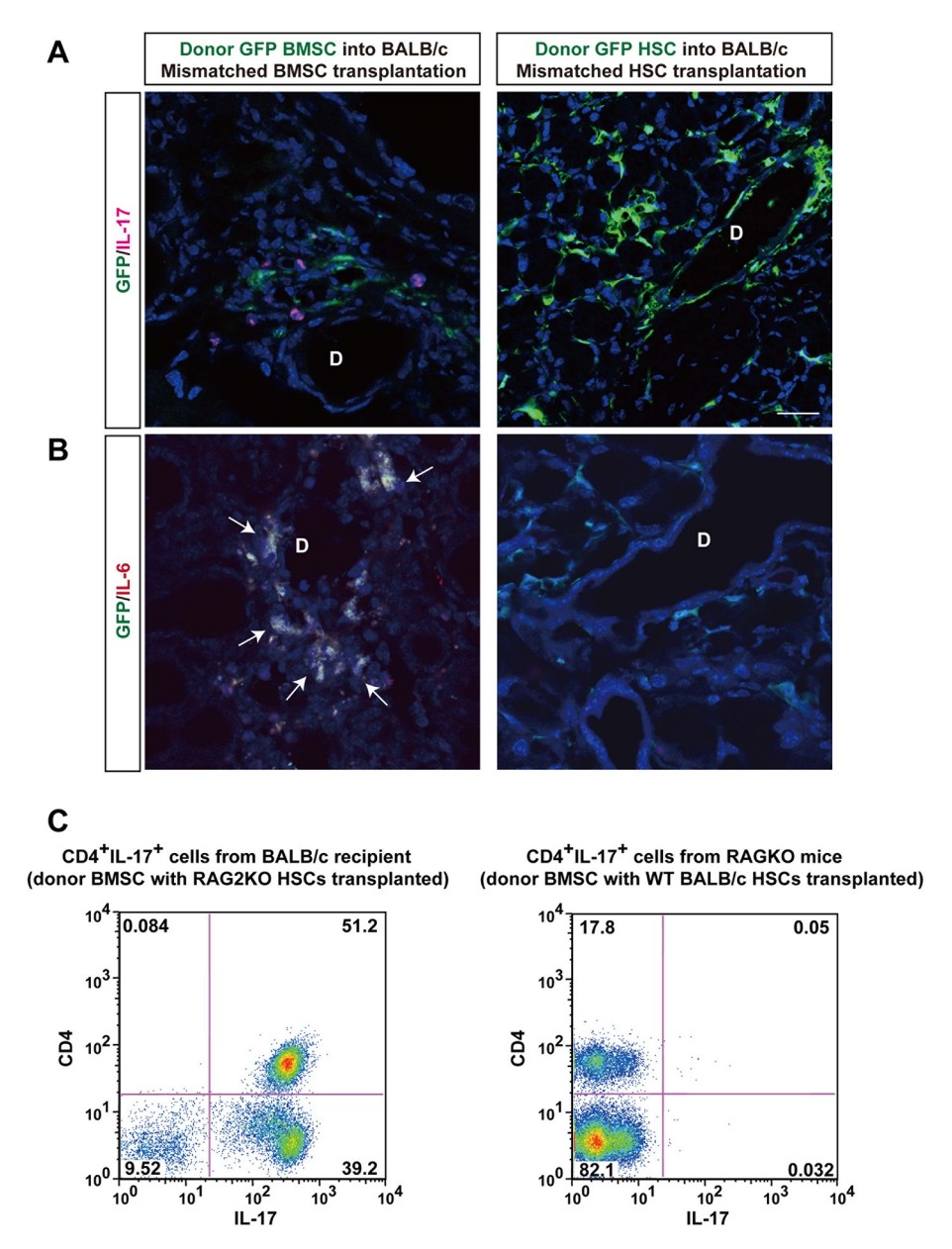

**Figure 4.** Donor BMSCs interact with recipient Th 17 cells and produce IL-6. (**A, B**) Th 17 cells (pink in **A**) and IL-6-producing cells (yellow in left panel of *Figure 4B*, arrows) were observed in the lacrimal gland of mismatched BMSC transplanted mice 8 weeks after transplantation. Yellow cells in (**B**) are due to co-localization of IL-6 (red) and donor BMSCs (GFP), resulting in yellow. Mismatched HSC transplanted mice did not show co-localization of donor cells (green) with IL-6 nor IL-17-producing cells. Representative images from two replicate experiments (n = 3 per group). Scale bar, 20 μm. (**C**) CD4$^+$IL-17$^+$ cells comprised more than 50% of splenic cells from B10.D2 BMSC + RAG2KO HSCs transplanted BALB/c recipients (left), while the ratio was low in spleens from B10.D2 BMSC + BALB/c HSC transplanted RAG2KO recipients (right). BMSC, bone marrow stromal/stem cells; HSC, hematopoietic stem cells.

The following figure supplement is available for figure 4:

**Figure supplement 1.** GFP donor MSCs produce IL-6.

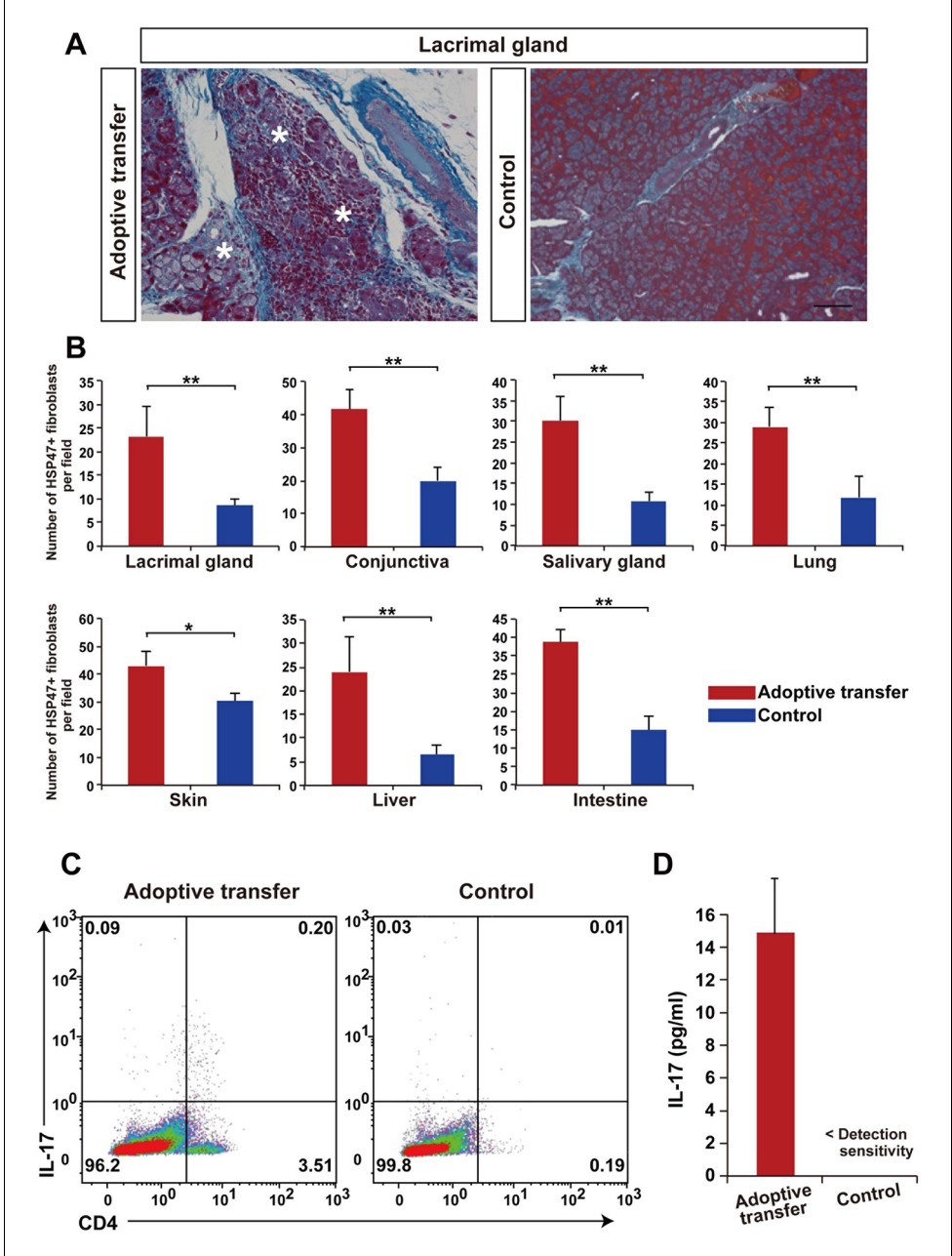

**Figure 5.** Adoptive transfer of recipient T cells from mismatched BMSC- transplanted recipients into BALB/c background Nude mice induce disease. (**A**) Adoptive transfer of BALB/c T cells from mismatched BMSC-transplanted mice induced excessive fibrosis accompanied by numerous inflammatory cells in the lacrimal gland of naive nude mice, as shown by Mallory staining (excessive fibrotic area in deep blue [*]). Scale bar, 100 μm. Data collected from two replicate experiments (n = 4 per group). (**B**) The number of HSP47[+] fibroblasts was significantly higher in various target organs following adoptive transfer of BALB/c T cells from mismatched BMSC recipients into nude mice (red), compared to wild-type (WT) nude mice (blue). Data are shown as mean ± SD. *p<0.005, **p<0.001. (**C**) CD4[+] Th 17[+] splenic cells were markedly elevated from adoptively transferred nude mice (left), compared to WT BALB/c background nude mice (right) (n = 4 each). (**D**) 1L-17 concentration in the serum from the same mice of the experiments shown in *Figure 4C* (n = 4 each). Data from one of two independent experiments (**A,D**). BMSC, bone marrow stromal/stem cells; HSP, heat-shock protein.

The following source data and figure supplement are available for figure 5:

**Source data 1.** Number of HSP47[+] cells in various target organs following adoptive transfer of BALB/c T cells from mismatched BMSC recipients into nude mice.

*Figure 5 continued on next page*

*Figure 5 continued*

**Source data 2.** 1L-17 concentration in the serum from adoptively transferred nude mice, compared to WT BALB/c background nude mice.
**Figure supplement 1.** Adoptive transfer of recipient T cells from mismatched BMSC-transplanted recipients into BALB/c background Nude mice induce disease in target organs other than the lacrimal glands.

## Activated T-cells in mismatched BMSC recipients

To confirm that mature host T cells react to mismatched BMSCs, Thy-1[+] T cells from various recipients (mismatched BMSC-transplanted, syngeneic BMSC-transplanted, untreated B10.D2 or untreated BALB/c mice) were co-cultured with freshly isolated PαS-BMSCs or splenic dendritic cells (DCs) from BALB/c or B10.D2 mice. T cells derived from mismatched BMSC-transplanted recipients proliferated and produced IL-6 in response to PαS-BMSCs but not DCs (*Figure 6A,B*). Interestingly, T cells also responded to matched (BALB/c) BMSCs (*Figure 6A,B*), suggesting that recipient T cells acquired the auto-reactive nature via antigen spreading (*Shlomchik, 2007*). In addition, either strain-derived BMSCs induced slight IL-6 secretion when reacted with naive T cells from wild-type mice (*Figure 6B*). Both PαS-BMSCs and T cells from mismatched BMSC-transplanted recipients produce IL-6 (*Figure 6C*).

Since the reaction was significantly suppressed by anti-MHC-class II antibody (*Figure 6D*) and CD4[+] T cells were predominant over CD8[+] T cells (*Figure 6E*), it was suggested that PαS-BMSCs themselves act as antigen-presenting cells via MHC-class II. Flowcytometry also revealed that MHC-class II molecule was expressed in only a rare subset of PαS cells, but the frequency of MHC-class II expressing cells was significantly increased after co-culture with recipient T cells (*Figure 6F*). These data suggest that mismatched BMSCs and host T cells stimulate each other via unknown but specific antigens expressed in mismatched BMSCs, but not in DCs, presented by MHC class II molecules.

## Temporal increase of regulatory T-cells is abrogated with mismatched BMSC

We further examined the in vivodynamics of T cells and BMSCs. We found that serum levels of IL-6 increased in recipient mice transplanted with mismatched BMSCs starting at 3 weeks after transplantation (*Figure 7A*). This coincided with the appearance of PαS-BMSC-derived cells in the peripheral blood of recipients starting at 3 weeks after transplantation and gradually increasing up to 7 weeks (*Figure 7B*). Interestingly, significant number of the PαS-BMSC-derived cells in the peripheral blood expressed MHC class II antigens, whereas the vast majority of them were negative prior to transplantation (*Figure 6F,7B*).

Increased IL-6 during an autoimmune process leads to the induction of Th17 cells and a decrease in regulatory T cells (Tregs) (*Bettelli et al., 2006*). We found that CD4[+]CD25[+]Foxp3[+] Tregs in peripheral blood transiently increased following lethal irradiation after syngeneic transplantation of BMSCs or WBM, which recovered to basal levels by 8 weeks. This transient increase in Tregs, reaching statistical significance at 3 weeks, was not observed after mismatched BMSC or WBM transplantation (*Figure 7C,D*). This was followed by an increase of Th17 cells in both the mismatched WBMT and BMSC groups reaching statistical significance compared to syngeneic controls at 3 weeks in the WBMT group, and 8 weeks in the BMSC mismatched group. Mismatched BMT more rapidly induces the appearance of CD4[+]/Th17[+] cells compared with mismatched PαS cells. (*Figure 7E,F*). These results suggest that the interaction of mismatched BMSCs and recipient T cells induce IL-6 production and lead to pathological changes in major organs similar to autoimmune disease, with inflammation due to Th17 cells via suppression of transient increase in Tregs.

## Discussion

We have reported the role of donor BMSCs in the pathogenesis of fibrosis associated with autoimmune SSc in a MHC-matched, minor antigen mismatched mouse model. Our data show that depletion of BMSCs from donor whole bone marrow significantly reduced fibrosis in all organs examined, and rescued mice from lacrimal gland dysfunction associated with the disease. These findings suggest the possible role of donor BMSCs in the initiation of the autoimmune process, as the number of

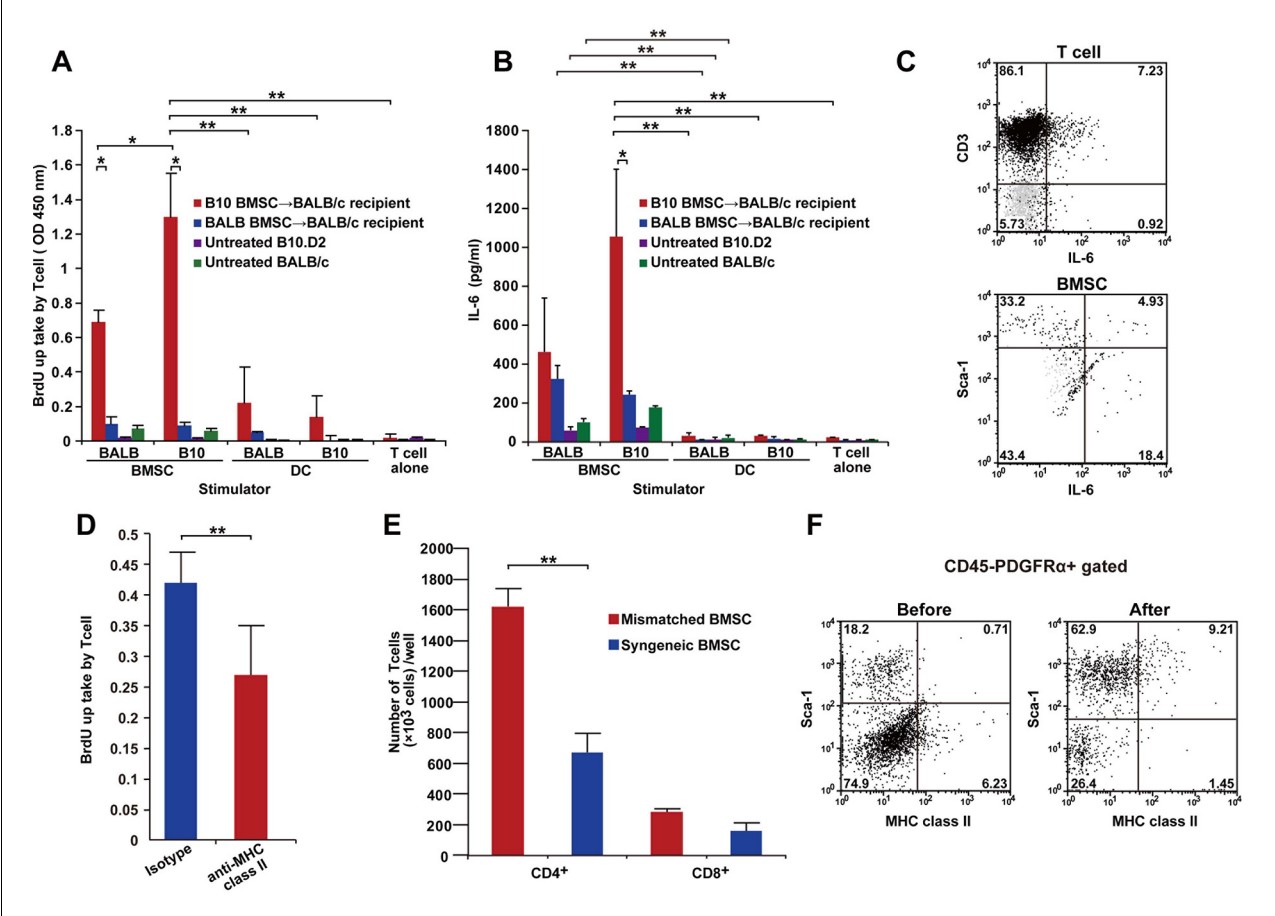

**Figure 6.** T cells following mismatched BMT are activated by PαS-BMSCs. (**A**, **B**) T cells isolated from mismatched BMSC-transplanted recipients proliferated when co-cultured with donor BMSCs (**A**), which was significantly blocked by anti-MHC class II antibody treatment (**D**). (**B**) Increased IL-6 production was observed following co-culture of T cells from mismatched BMSC-transplanted recipients with donor PαS-BMSCs, but not with splenic dendritic cells (DCs). Color bars indicate source of T cells. Results are from triplicate cultures of two independent experiments in both (**A**) and (**B**) and from quintupulicate of two independent experiments in (**D**). Data are shown as mean ± SD. *p<0.05, **p<0.01. (**C**) Both CD3[+] T cells and Sca-1[+] BMSCs produced IL-6 following co-culture described in (**A**). Dot plots of mismatched BMSCs-transplanted recipient samples are shown in black, and isotype control in light grey. (**E**) The increase in T cell proliferation under co-culture with mismatched BMSCs (red) was due to the activation of CD4[+] and not CD8[+] T cells. Data collected from triplicate cultures of two replicate experiments (n = 2 per group). Data are shown as mean ± SD. **p<0.01. (**F**) MHC class II expression was upregulated in BMSCs after co-culture with T cells from mismatched BMSC-transplanted recipients. BMT, bone marrow transplantation; BMSC, bone marrow stromal/stem cells; SD, standard deviation.

The following source data is available for figure 6:

**Source data 1.** T cell proliferation after co-culturing of donor or recipient BMSCs and splenic dendritic cells (DC).

**Source data 2.** IL-6 production following co-culture of T cells from various sources with donor or recipient BMSCs and splenic dendritic cells (DCs).

**Source data 3.** T cells proliferation blocked by anti-MHC class II antibody treatment.

**Source data 4.** CD4[+] T cells and CD8[+]T cells proliferation under co-culture with syngeneic or mismatched BMSCs.

BMSCs in graft correlated with severity of fibrosis (*Figure 1E,F*). Furthermore, the onset of true auto-immune disease in these mice was shown by the repression of Treg induction, followed by an increase in Th17 effector cells of host origin. This shows that migrating donor PαS-BMSCs are the initial trigger of events leading to increased levels of circulating IL-6, followed by a decrease in Tregs, and conversely an increase in host-derived Th17 cells. Activation or maturation of BMSC-derived

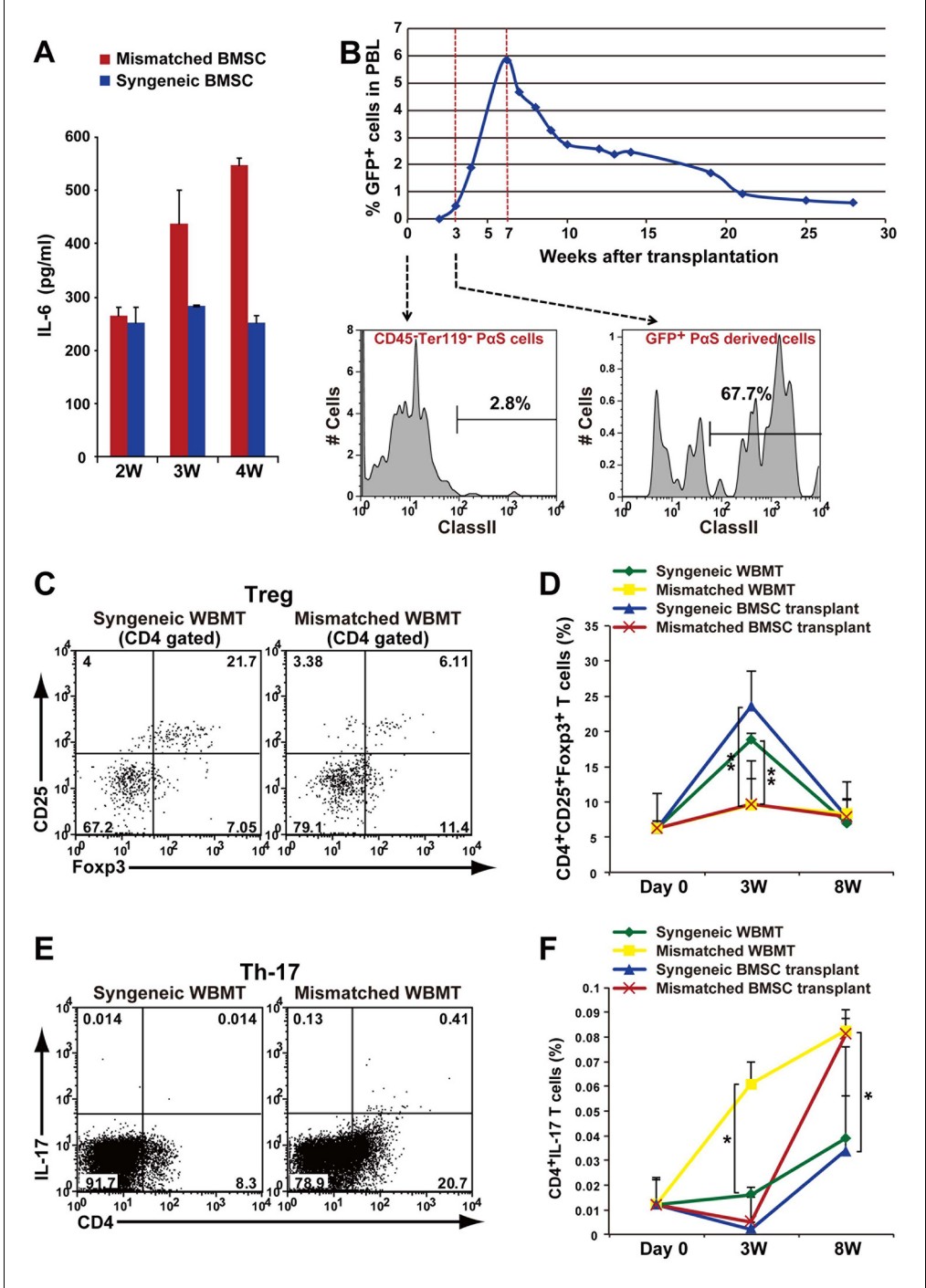

**Figure 7.** Autoimmune phenotype following mismatched BMSC transplantation. (A) Serum IL-6 concentration increased after mismatched BMSC transplantation (red) compared to syngeneic control (blue). Duplicate experiments. Data are shown as mean ± SD, n = 2. (B) GFP[+] donor BMSCs appear in peripheral blood mononuclear cells after mismatched BMSCs transplantation, and peaks at approximately 7 weeks. The percentage of BMSCs expressing MHC class II antigen increases following transplantation. (C) CD4[+] CD25[+] Foxp3[+] Tregs were suppressed in the spleen after mismatched WBMT transplantation (right) compared to syngeneic control (left). (D) CD4[+]CD25[+]Foxp3[+] Tregs were suppressed in both mismatched WBMT (yellow) and mismatched BMSC transplantation (red) compared to syngeneic WBMT (green) or syngeneic BMSCs transplantation (blue). Data are shown as mean ± SD, triplicate experiments, n = 3–5, **p<0.01. (E, F) The ratio of CD4[+] IL-17[+] T cells in the spleen was significantly higher following mismatched WBMT (yellow) or mismatched BMSC transplantation (red) compared to syngeneic control (green and blue). Data are shown as mean ± SD, triplicate experiments, n = 3–5, *p<0.05. BMSCs, bone marrow stromal/stem cells; SD, standard deviation.

*Figure 7 continued on next page*

*Figure 7 continued*

The following source data is available for figure 7:

**Source data 1.** Serum IL-6 concentration after mismatched BMSC transplantation compared to syngeneic BMSC transplantation.
**Source data 2.** Serial changes of CD4$^+$CD25$^+$Foxp3$^+$ Tregs in spleen cells.
**Source data 3.** The ratio of CD4$^+$ IL-17$^+$ T cells in the spleen cells.

progeny is probably also involved in the progression of disease because accumulation of donor-derived fibroblasts was not observed in syngeneic BMSC transplantation, where fibrosis does not occur. In the mixed lymphocyte reaction experiments, Thy-1$^+$ T-cells isolated from host mice after mismatched BMT were activated by PαS-BMSCs in vitro, as shown by enhanced proliferation and IL-6 secretion. In addition, the T-cell proliferation was blocked by anti-MHC antibody. Finally, adoptively transferred T cells from mismatched BMSCs recipients into nude-mice-induced autoimmune-like inflammation and fibrosis in targets organs, indicating that autoreactive recipient T cells were activated by antigens presented by MHC class II molecules in minor antigen mismatched donor BMSCs. Adoptive transferred T cells react recipient BMSCs leading to recipient' fibrosis (*Figure 5*) probably because adoptively transferred T cells has already acquired the autoreactive nature to activate the recipient derived BMSCs as shown in vitro analysis from *Figure 6A* and *Figure 6B*.

Although the auto-antigen responsible for the autoimmune type reaction still needs to be identified, a common antigen specifically expressed in BMSCs of B10.D2 and BALB/c with different isoforms or SNIPs, is one of the most probable candidates. Since naive T cells responded to PαS-BMSCs in vitro at low levels, we speculate that auto-reactive T cells that are only present in basal levels under normal conditions (*Sakaguchi et al., 2008*) can recognize minor differences between B10.D2 and BALB/c BMSCs. The lack of fibrosis following mismatched HSC transplantation suggests that hematopoietic lineage cells, including DCs, did not express such a molecule. The identical phenotype was observed when host and recipient were reversed (i.e. BALB/c BMSCs transplanted to B10.D2 recipients, data not shown). This implies that both B10.D2 and BALB/c BMSCs can be a primary inducer against mismatched recipient T cells, showing that the autoimmune-related fibrosis in this model was due to difference in strain, and not a specific reaction of B10.D2 BMSCs and BALB/c T cells.

The novelty of our study is the use of prospectively isolated BMSCs combined with prospectively isolated HSCs. Previous studies have used culture-isolated BMSCs, which do not reflect the physiological role of these cells in vivo. Functional differences between in vitro and in vivo observations may explain discrepancies in the anti-inflammatory and pro-inflammatory effects of BMSCs reported in the literature (See review by Bernardo et. al. (*Bernardo and Fibbe, 2013*). Our study has also demonstrated that donor BMSCs can contribute to the fibroblast population observed in fibrotic lesions of the host. Our findings suggest that once mismatched BMSCs migrate into the target organs via certain homing signals, mismatched BMSCs encounter T cells and proliferate and activate under the pathological microenvironment (*Figure 2A*). This phenomenon may not be present in syngeneic BMSCs transplantation where an allogeneic response does not occur. As for HSP47 expression, it is clearly seen in activated fibroblasts, but only very faint expression is detected in quiescent fibroblasts in the syngeneic BMSC-transplanted recipients' target organs. Interestingly, a fraction of BMSCs mobilized in the peripheral blood expressed MHC class II molecules (*Figure 6B*). Preliminary data show that these cells also express CD45 and type I collagen (data not shown), which corresponds with the phenotype of fibrocytes reported in the literature (*Abe et al., 2001*; *Chesney et al., 1997*; *Mielcarek et al., 2003*; *Phillips et al., 2004*; *Wang et al., 2007*; *Yang et al., 2002*). Further studies are required to elucidate the association of PαS-BMSCs and fibrocytes.

Interestingly, this SSc mouse model has also been used as a model of chronic graft-versus-host disease (cGVHD) (*Kaplan et al., 2004*; *Kim et al., 2007*). The phenotype of cGVHD characterized by systemic fibrosis and severe dry eye is very similar to this SSc mouse phenotype. Unlike conventional cGVHD that is believed to be caused by donor T cells, our results show that radio-resistant residual recipient T cells, but not donor-HSC derived de novo T cells, were activated following mismatched BMSC transplantation. Although we do not believe that our findings in this mouse model could be

directly applied to human cGVHD cases, it is worth noting this phenomenon. In fact, several previous reports have shown that residual recipient CD4[+] T cells regulate cGVHD(*Anderson et al., 2004*; *Blazar et al., 2000*; *Jaffee and Claman, 1983*). In addition, cGVHD has occurred in a surprisingly high fraction of nonmyeloablative stem cell transplant recipients (*Anderson et al., 2004*; *Mielcarek et al., 2003*; *Schetelig et al., 2002*) since residual host T cells remain in nonmyeloablative transplantation.

Notably, BMSC depletion from grafts significantly reduced fibrosis in all organs that we examined. The use of CD34[+] selected HSCs as an approach to reducing the risk of cGVHD was suggested in early studies of CD34[+] selected peripheral blood HSCs. (*Martínez et al., 1999*; *Urbano-Ispizua et al., 1997*; *2001*) CD34[+] selected HSC transplantation depletes not only mature T cells but also donor BMSCs because human BMSCs are negative for CD34 expression (*Mabuchi et al., 2013*). In addition, recent reports showed that the incidence of cGVHD was less following cord blood transplantation (CBT) compared to peripheral blood stem-cell transplantation (PBSCT) (*Takahashi et al., 2007*; *Uchino et al., 2012*). We have found that BMSCs are rarely detected in cord blood but are abundant in G-CSF-mobilized PBL (*Mabuchi et al., 2013*), which may indicate that the onset of cGVHD may correlate with the number of BMSCs transplanted. Taken all, the reduced risk of cGVHD may have been due to the removal or absence of donor BMSCs as we showed in our BMSC-depleted model. Further studies are required for monitoring the frequency of donor BMSCs in grafts and residual host T cells in human cGVHD patients to elucidate the possible role of donor BMSCs in the pathogenesis.

In summary, prospective transplantation of freshly purified BMSCs and HSCs into BALB/c-RAG2KO suggests that transplantation of minor antigen-mismatched MHC-compatible BMSCs interact with residual host T cells to induce the autoimmune phenotype observed in fibrosis associated with the SSc mouse model. While the responsible antigen remains to be elucidated, our data suggest that accidental recognition of self-minor antigens on MHC class II[+] BMSCs may be involved in systemic fibrosis observed in autoimmune disease.

## Materials and methods

### Mice
B10.D2, BALB/c, and BALB/c background Nude mice 8-10 week of age were purchased from Sankyo Laboratory, Inc. (Tokyo, Japan). B10.D2 GFP mice were obtained by backcrossing of B10.D2 mice with C57BL/6 GFP mice (Japan SLC Ltd). The 10th generation of backcrossed B10.D2 GFP progenies was used for experiments. RAG2KO on BALB/c background were kindly provided from Dr. Shigeo Koyasu (Keio University). All experimental procedures and protocols were approved by the ethics committee of Keio University and were in accordance with the Guide for the Care and Use of Laboratory Animals (# 09152). We followed the ARRIVE guidelines for reporting in vivo experiments in animal research (*Kilkenny et al., 2010*).

### Isolation of SP-HSCs
Bone marrow cells, suspended at 1x10[6] cells/ml in calcium- and magnesium-free Hanks balanced salt solution (HBSS) supplemented with 2% fetal calf serum, 10 mM HEPES, and 1% penicillin/streptomycin (HBSS+) were incubated with 5 μg/ml Hoechst 33342 (Sigma Aldrich, St. Louis, MO) for 60 min at 37°C. The side population (SP) was sorted as described previously (*Matsuzaki et al., 2004*). (*Figure 1—figure supplement 2*)

### Isolation of CD45[-]TER119[-]PDGFRα[+]Sca-1[+]cells (BMSCs)
Purified BMSCs were isolated by flowcytometry as described previously (*Morikawa et al., 2009a*; *2009b*). Briefly, femurs and tibias were dissected and crushed with a pestle. The bone marrow was gently washed in HBSS+. Collagenase (0.2%) was used to digest the minced tibia and femur in DMEM without fetal bovine serum at 37°C for 1 hr. After red blood cell lysis, the residual cells were stained with FITC-labeled anti-Sca-1 (Ly6A/E), APC-labeled anti-PDGFR-α (APA-5), PE-labeled anti-CD45 (30-F-11), and PE-labeled anti-TER119 (TER-119) (all from e-Bioscience, San Diego, CA). Analysis and sorting were performed on a triple-laser MoFlo flow cytometer (Beckman Coulter, Brea, CA). CD45[-] TER119[-] PDGFRα[+] Sca-1[+] cells were routinely prepared at 99% purity by this method

(*Figure 1—figure supplement 2*) (*Morikawa et al., 2009a*). CD45[-] TER119[-] PDGFRα[+] Sca-1[+] cells were further characterized by using FITC-labeled anti-Sca-1 (E13-161.7), PE- or APC-labeled anti-PDGFR-α (APA-5), PE-Cy7-labeled anti-CD45 (30-F-11), and PE-Cy7 labeled anti-TER119 (TER-119) (all from BD Pharmingen, San Jose, CA) and then Sca-1 and PDGFR-α double positive fraction were further analyzed with PE-labeled anti-CD31 (MEC13.3: BD Pharmingen), PE-labeled anti-CD133 (13A4: e-Bioscience), PE-labeled anti-CD 29 (Ha2/5: BD Pharmingen), APC-labeled anti-CD90 (30-H12: BioLegend, San Diego, CA), and APC-labeled CD106 (429: BioLegend), respectively. For PE-labeled isotype antibody, rat IgG2b, κ for CD45 and TER119 (e-Bioscience), and rat IgG2a, κ for CD31 (BD phamingen), rat IgG1, κ for CD133 (BD phamingen), and Armenian Hamster IgM, κ for CD29 (BD phamingen) were used. For APC-labeled isotype antibody, rat IgG2a, κ for, rat IgG2b, κ for CD90 (BD phamingen) and rat IgG2a, κ for PDGFRα[+] (e-Bioscience) and CD106 (BD phamingen) were used. FITC-labeled rat anti-mouse IgG2a, κ for Sca-1(e-Bioscience) was used.

## BMSCs and HSCs co-transplantation

For co-transplantation experiments, $1\times10^4$ PαS-BMSCs and $1\times10^3$ SP-HSCs (see 'Methods' and *Figure 1—figure supplement 2*) were intravenously injected into the tail vein (200 µl/ per mouse) of recipient mice that had been lethally irradiated with a dose of 7.0 Gy as indicated in our previous studies (*Matsuzaki et al., 2004*; *Morikawa et al., 2009a*).

This is approximately equivalent to 50 CFU-F in $1\times10^4$ PDGFRα[+]/Sca1[+] cells (*Morikawa et al., 2009a*; *2009b*) while approximately 10 to 50 HSCs are found in 1000 SP KSL cells (*Okada et al., 1992*). Therefore, while the ratio of PαS-BMSC to SP KSL was 10:1 in the donor, the actual number of BMSCs to HSCs was approximately 1:1 (*Okada et al., 1992*).

## T-cell isolation for co-culture with BMSCs

Spleens were obtained from mice that received B10.D2 BMSC or BALB/c BMSC transplants. T cells and DCs were purified from the spleens by anti-CD90.2 monoclonal antibody (mAb)-conjugated microbeads or anti-CD11c mAb-conjugated microbeads (Miltenyi Biotic, Bergisch Gladbach, Germany), respectively, according to the manufacturer's instructions. The purity was consistently > 98%. T cells were cultured alone or co-cultured with PαS-BMSCs at a ratio of 10:1 (T cells: BMSCs).

## Adoptive transfer of splenic T cells from mismatched BMSC transplanted recipients.

Adoptive transfer was performed in accordance with previous publication (*Arakaki et al., 2003*; *Niederkorn et al., 2006*). Briefly, splenic T cells were isolated from mismatched PαS-BMSC transplanted BALB/c recipients as described above, and $3\times10^6$ cells were transferred into BALB/c background nude mice without any pre-treatment. The animals were sacrificed 45 days after transfer. Analyses of tissue samples from target organs were performed as shown in the section 'Immunohistochemistry and immunofluorescence'. Splenic cells were collected and prepared for flowcytometry and stained with CD4and IL-17 antibody as shown in the section 'Flowcytometry analysis for IL-17'.

## Proliferation assay

Purified B10.D2 PαS-BMSCs and BALB/c PαS-BMSCs were plated at $1\times10^4$ cells/ well into 96-well plates in triplicate, which were irradiated at 52 Gy after adherence. $1\times10^5$ purified mouse T cells were added to each well. On the fourth day, 5-bromo-2'-deoxyuridine (BrdU) was added. Twenty-four hours later, BrdU uptake was quantified by cell proliferation enzyme-linked immunosorbent assay (ELISA) using a BrdU Kit (Roche Applied Science, Penzberg, Germany) (*Guo et al., 2009*). In some experiments, purified B10.D2 PαS-BMSCs were co-cultured with T cells isolated from mismatched PαS-BMSC-transplanted recipient mice. Cells were treated with either blocking anti-MHC class II antibody (M5/114.15.2, Biolegend) or isotype control. The supernatant of the co-cultures was subjected to IL-6 ELISA using commercial kit (BD Biosciences).

## Flowcytometry analysis for Foxp3 and intracellular cytokine IL-6 and IL-17

For Foxp3 staining, whole blood samples were co-stained with anti-CD4-FITC and anti-CD25-PE (PC61.5). After fixation and permealization, cells were stained with APC-labeled anti-Foxp3 mAb (FJK-16s) (all mAbs were from e-Bioscience) as described (*Chen et al., 2007*). For IL-6, and IL-17 staining, $2 \times 10^6$ spleen cells were stimulated with 10 ng/ml phorbol myristate acetate (PMA) (Sigma) and 10 ng/ml ionomycin (Sigma) in the presence of the Golgi inhibitor Brefeldin (Sigma) (10 µg/ml) for 4 hr. The cells were then stained with FITC-labeled anti-CD4 (GK 1.5, e-Bioscience) and PE-labeled anti-IL-6 (MP5-20F3, BD Pharmingen) or APC-labeled anti-IL-17 (eBio17B7, e-Bioscience) as described (*Kappel et al., 2009*). FITC-labeled anti-CD8 (53-6.7) and MHC class II (M5/114.15.2, e-Bioscience) were used for FACS analysis on T cells and BMSCs. PE-labeled IgG2a, FITC-labeled IgG1, APC-labeled IgG2a and IgG2a, κ were used as isotype controls (all from e-Bioscience). Cells were analyzed on a FACScan with Cellquest software (Beckton Dickinson). For the co-cultures with T cells and BMSCs, T cells were stimulated with Brefeldin (10 µg/ml) alone for 4 hr.

## Statistics

Bonferroni/Dunn test (SPSS19.0 for Windows, SPSS Japan Inc., Tokyo, Japan) and two tailed Stutent's t-test was used to analyze the number of HSP47+ fibroblasts per field. Two-tailed Student's t-test (SPSS19.0 for Windows) was used to analyze tear volume, cytokine serum levels, and cell proliferation and cytokine production in co-cultures. Differences were considered significant when $p < 0.05$. Data presented as mean $\pm$ SD.

## Supplemental methods

### Whole bone-marrow transplantation

In a typical WBM transplantation, we used 7- to 8-week-old male B10.D2 (H-2d) and female BALB/c (H-2d) mice as donors and recipients, respectively. Briefly, $1 \times 10^6$ whole bone marrow cells and $2 \times 10^6$ spleen cells (200 µl total) were intravenously injected into the tail vein of recipient mice that had been lethally irradiated with a dose of 7.0 Gy, as indicated in a previous study (*Zhang et al., 2002*). Three to five animals per group were studied 3 and 8 weeks after transplantation for each experiment.

### Immunohistochemistry and immunofluorescence

Animals were anesthetized using anesthetics by intraperitoneal injection of 5 mg/ml sodium pentobarbital (Sumitomo Dainippon Pharma Co., Ltd, Osaka, Japan) and fixated by perfusion with 4% paraformaldehyde (PFA) at 3 and 8 weeks after transplantation. For histologic examinations, the lacrimal gland, eye, salivary gland, lung, skin, intestine, liver, and thymus were excised, fixed in 4% PFA overnight, embedded in paraffin wax, and processed for hematoxylin-eosin and Mallory staining. Tissue sections of target organs subjected to Mallory staining were assessed for morphometric analysis. A minimum of five randomly selected fields were captured at 200X magnification for each section using a Nikon Coolsope II (Nikon). The fibrotic areas was quantified as the ratio of the blue-stained area to the total stained area (*Brack et al., 2007*).

Immunohistochemistry was performed on formalin-fixed paraffin-embedded or frozen tissue sections. The staining patterns were graded semi quantitatively according to the intensity and distribution of the labeling, as described previously (*Ogawa et al., 2009*). For immunofluorescent staining for HSP47, a collagen-specific molecular chaperon (SPA-470, Stress Gen Biotechnologies Corp), antigen unmasking was done for paraffin-embedded sections by autoclaving at 120°C for 20 min or incubating the sections at 37°C in antigen retrieval solution (Histo VT One, Nacalai Tesque) for 10 min for formalin fixed-frozen sections. For frozen sections, the sections were then blocked with 10% goat serum for 30 min and incubated overnight at 4°C with an anti- PE labeled rat anti- mouse IL-6 antibody and APC labeled anti-mouse IL-17 antibody. HSP47 antibody cross-reacts with the mouse HSP47 antigen. After washing with PBS, the sections were incubated with an Alexa 488-conjugated goat anti-mouse secondary antibody (Molecular Probes) and the nuclei were stained simultaneously with DAPI. Isotype matched mouse antibodies including mouse Ig G 2b, κ antibody for HSP47, rat Ig G1 antibody for IL-6 and rat IgG2a,κ antibody for 1L-17 were prepared for negative control. Tissue sections for fluorescent staining were examined with an LSM 700 confocal microscope (Carl Zeiss,

Jena, Germany). To assess the histological architecture and staining, all the acquired images were reviewed twice each by two independent observers who were blinded to the source of the samples. The number of HSP47$^+$ fibroblasts/field (at x630 magnification) was counted in at least five different fields on three to five sections. HSP47$^+$ spindle-shaped cells with an oval nucleus that resided in the interstitium were regarded as fibroblasts (*Ogawa et al., 2005*).

### Serum cytokine analysis

Serum was collected from mice using retro-orbital beads (*Chen et al., 2009*). The concentration of IL-6 and IL-17 in the supernatants was determined by ELISA using a specific kit, according to the manufacturer's instructions (BD Biosciences for IL-6, and R&D systems for IL-17). The experiments were performed in triplicate.

### Tear secretion volume measurements

Fibrosis-induced and control mice were anesthetized by intra-peritoneal injection of 5 mg/ml sodium pentobarbital (Sumitomo Dainippon Pharma). Tear secretion was evaluated as described (*Ma et al., 1999*;*Inaba et al., 2014*). Animals were stimulated 3 min after anesthesia by intra-peritoneal injection of 0.02% pilocarpine solution (1 mg/kg; Santen Pharmaceutical Co., LTD), and tears were collected from the conjunctival sac of the lateral canthus for 15 min using 0.5-µl diameter capillary microglass tubes (Microcaps, Drummond Scientific Company, Broomall, PA). The total tear volume was determined as the average of three separate measurements of 5 min each.

## Acknowledgements

The authors would like to thank Dr. Shigeo Koyasu for providing the RAG2 KO mice, Dr. Masataka Kuwana for critical advice, Dr. Takashi Kobayashi for valuable discussion and proof reading of the manuscript, and Ms. Mai Tadaki for technical assistance.

## Additional information

### Competing interests

HO: Reviewing editor, *eLife.* The other authors declare that no competing interests exist.

### Funding

| Funder | Author |
| --- | --- |
| Japanese Ministry of Education, Science, Sports and Culture | Yoko Ogawa |
| Japan Society for the Promotion of Science | Yumi Matsuzaki |
| Global Century COE program of the Ministry of Education | Yumi Matsuzaki Shigeto Shimmura |
| Japan Women Medical Association | Yoko Ogawa |
| Japan Medical Association | Yoko Ogawa |

The funders had no role in study design, data collection and interpretation, or the decision to submit the work for publication.

### Author contributions

YO, SMo, Conception and design, Acquisition of data, Analysis and interpretation of data, Drafting or revising the article, Contributed unpublished essential data or reagents; Performed experiments; Approved the final version of the manuscript; HO, KT, SSh, Conception and design, Analysis and interpretation of data, Drafting or revising the article, Contributed unpublished essential data or reagents; Approved the final version of the manuscript; YMa, SSu, TY, YS, SY, TI, Acquisition of data, Analysis and interpretation of data, Contributed unpublished essential data or reagents;

Performed experiments; Approved the final version of the manuscript; SMu, Acquisition of data, Analysis and interpretation of data; Performed experiments; Approved the final version of the manuscript; SO, Acquisition of data, Analysis and interpretation of data, Contributed unpublished essential data or reagents; Approved the final version of the manuscript; YK, Acquisition of data, Analysis and interpretation of data, Contributed unpublished essential data or reagents; Drafted and revised the manuscript; Approved the final version of the manuscript; YMa, Conception and design, Acquisition of data, Analysis and interpretation of data, Drafting or revising the article, Contributed unpublished essential data or reagents; Approved the final version of the manuscript

### Ethics

Animal experimentation: All experimental procedures and protocols were approved by the ethics committee of Keio University and were in accordance with the Guide for the Care and Use of Laboratory Animals (# 09152). We followed the ARRIVE guidelines for reporting in vivo experiments in animal research (Kilkenny, et al., 2010).

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
