## [Decision Letter]

Thank you for submitting your work entitled "MHC-compatible donor mesenchymal stem cells trigger fibrosis by activating host T cells in a scleroderma mouse model" for peer review at *eLife*. Your submission has been favorably evaluated by Tadatsugu Taniguchi (Senior editor), a Reviewing editor, and two reviewers.

The reviewers have discussed the reviews with one another and the Reviewing editor has drafted this decision to help you prepare a revised submission.

Summary:

This manuscript describes the possible importance of donor mesenchymal stem cells (MSCs) and host T cells in what is typically considered a murine cGVHD model. The authors report that mismatched MSCs are a key inducer of fibrosis after BMT and that host T cells, but not donor T cells, are required for the progression of this fibrosis, presumably as a result in a break in tolerance and the initiation of an autoimmune state. This study is potentially important for changing existing views of the pathogenesis of cGVHD. However, there are some doubts about the results, which have to be answered before publication.

Essential revisions:

While the authors have reported that freshly isolated PDGFRα^+^/Sca-1^+^ are able to survive after systemic delivery and possibly to form differentiated cell types, the vast majority of studies report that BMSCs are not transplantable. There is related to the question of how "pure" the PαS cells are. It is not at all clear that PαS "MSCs" are as uniform a population of cells as the authors think. While they do have features of the best-characterized "MSC", bone marrow skeletal stem cells, they have other properties that may be due to the inclusion of other cell types using their minimal cell surface phenotype. As reported in Morikawa et al, 2009a, endothelial cells are Sca-1^+^ (and it has been reported that endothelial progenitors express PDGFRα). Collagenase was used to digest the marrow prior to sorting, thereby liberating more cells that the usual mechanically dissociation. This suggests that endothelial and or progenitor cells are likely to be within the PαS fraction and indeed, the statements about MHC class II+ fibrocytes suggest that even hematopoietic cells may contaminate the injected population. Further, it is stated that "PαS-MSCs" were found in peripheral blood, but to date there is rigorous evidence that bona fide BMSCs are able to circulate on a routine basis. This calls into question the efficacy of the markers used (PDGFRα and Sca-1) to identify only BMSCs. The authors need to show a better characterization of the PαS cells (CD45^-^/Ter119^-^/PDGFRα^+^/Sca-1^+^ cells), specifically, to show that they have other markers of bone marrow stromal cells (e.g., CD29, CD, 90, CD106) and devoid of endothelial cells (CD31), endothelial progenitors (CD133), or hematopoietic cells, especially of myeloid nature. This is key for the authors to claim that the fibrosis is induced by mismatched bone marrow stromal cells.

The authors show that almost all spleen cells were donor-derived in matched BMT, whereas the majority of them were host-derived cells after mismatched BMT using immunofluorescence (Figure 2). However, this result is not enough to establish this point. The authors should show the flow cytometry results, using GFP or anti Ly-9.1 antibody. Since anti Ly-9.1 antibody react with lymphocyte of BALB/c mice but not B10.D2, this antibody is useful for distinguishing host cells from donor cells.

The authors show a significant increase of recipient derived CD4^+^Th17 PBMC cells (Figure 4, Left). If this result is true, all of CD4^+^ T cells produce high levels of IL-17. Furthermore, almost all of other cells also produce high levels of IL-17. Does it increase serum IL-17 levels in this mouse?

Figure 1 – unlike what is stated in the text, and enumerated in Figure 1, there do appear to be fibrotic changes in the lacrimal gland (albeit minimal), and in the salivary gland and perhaps in the intestine when BALB/c mice received B10 HSCs. There also appears to be fibrosis in the lacrimal gland, salivary gland lungs, perhaps in liver and intestine in the mice receiving bone marrow depleted of PαS cells. Fibrotic lesions in all five groups were only analyzed in the lacrimal glands. It is stated that mismatched BMSC-induced fibrosis was dose dependent, but it only appears to be statistically significant when a dose that is 3.2x higher than the starting dose is used. The authors report using 10,000 BMSCs routinely, and this falls between the 8,000 and 16,000 cells used in Figure 1.

In Figure 2, colocalization of GFP and HSP47 cannot be well seen in the lacrimal gland and in the intestine. In 2C where the cells are syngeneic, very few GFP^+^ cells are seen in the lacrimal gland and none are seen in the skin and intestine, which makes one ask why the cells seem to survive in the mismatched experiments. Colocalization of GFP and HSP47 cannot be well seen in the in vitro culture of lacrimal fibroblasts.

Figure 3—figure supplement 1 is more rightly a supplement for Figure 2. In Figure 2, what is missing is the enumeration of spleen cells in the syngeneic and mismatched WBMT mice in order to say that the cells were recipient in nature.

In Figure 4, left panel, no GFP staining is visible, and IL-6 staining is very faint.

Figure 5 shows only the lacrimal gland (other organs are listed in the text). Unlike what is stated in the text (that CD4^+^/Th17^+^ cells were "markedly" elevated), Figure 5 shows that the increase in CD4^+^/Th17^+^ cells in the adoptively transferred recipients is very limited (0.2% compared to 51.2% shown in Figure 4 for mismatched BMSCs placed into BALB/c mice), yet the fibrosis after the adoptive transfer is marked. The authors need to provide an interpretation of this result.

In Figure 6, some important statistical comparisons are missing, and are needed to support the statement that IL-6 secretion was stimulated by BMSCs from either strain of mice. The data in Figure 6 should be reorganized to match the order of presentation in the text.

---

## [Author Response]

*While the authors have reported that freshly isolated PDGFRα^+^/Sca-1^+^ are able to survive after systemic delivery and possibly to form differentiated cell types, the vast majority of studies report that BMSCs are not transplantable. There is related to the question of how "pure" the PαS cells are. It is not at all clear that PαS "MSCs" are as uniform a population of cells as the authors think. While they do have features of the best-characterized "MSC", bone marrow skeletal stem cells, they have other properties that may be due to the inclusion of other cell types using their minimal cell surface phenotype. As reported in Morikawa et al, 2009a, endothelial cells are Sca-1^+^ (and it has been reported that endothelial progenitors express PDGFRα). Collagenase was used to digest the marrow prior to sorting, thereby liberating more cells that the usual mechanically dissociation. This suggests that endothelial and or progenitor cells are likely to be within the PαS fraction and indeed, the statements about MHC class II+ fibrocytes suggest that even hematopoietic cells may contaminate the injected population. Further, it is stated that "PαS-MSCs" were found in peripheral blood, but to date there is rigorous evidence that bona fide BMSCs are able to circulate on a routine basis. This calls into question the efficacy of the markers used (PDGFRα and Sca-1) to identify only BMSCs. The authors need to show a better characterization of the PαS cells (CD45^-^/Ter119^-^/PDGFRα^+^/Sca-1^+^ cells), specifically, to show that they have other markers of bone marrow stromal cells (e.g., CD29, CD, 90, CD106) and devoid of endothelial cells (CD31), endothelial progenitors (CD133), or hematopoietic cells, especially of myeloid nature. This is key for the authors to claim that the fibrosis is induced by mismatched bone marrow stromal cells.*

We appreciate the editors’ and reviewers’ constructive comments and suggestions. We further analyzed the characteristics of wild type B10.D2 PαS BMSCs. We confirmed that PαS BMSCs cells expressed CD29, CD90, and CD106 as shown in the new Figure 1—figure supplement 2. Almost all PαS cells were CD31, and CD133 negative, suggesting that endothelial cells and their progenitors contributed minimally to the population.

We added FACS analysis figures for MSC characterization in Figure 1—figure supplement 2 and described the characteristics of PαS BMSCs in the text and the figure legend as follows:

“We confirmed that PαS BMSCs cells expressed CD29, CD90, and CD106, known as established markers of BMSCs. However, CD31 and CD133, markers of endothelial cells, consisted of only a small portion of the PαS BMSCs fraction (Figure 1—figure supplement 2). These PαS-BMSCs and side population (SP-HSCs) from B10.D2 or BALB/c were systemically co-transplanted into BALB/c recipients as defined combinations (Figure 1)”.

We added the Figure 1—figure supplement 2.

We added figure legend for the Figure 1—figure supplement 2 as follows:

“(C) Characterization of PαS BMSCs by other BMSCs marker CD29, CD90 and CD106 and endothelial marker, CD31 and CD133 by flowcytometry.”

In the Methods as follows:

“CD45^-^ TER119^-^ PDGFRα^+^ Sca-1^+^ cells were further characterized by using FITC-labeled anti-Sca-1 (E13-161.7), PE- or APC-labeled anti-PDGFR-α (APA-5), PE-Cy7-labeled anti-CD45 (30-F-11), and PE-Cy7 labeled anti-TER119 (TER-119) (all from BD Pharmingen) and then Sca-1 and PDGFR-α double positive fraction were further analyzed with PE-labeled anti-CD31 (MEC13.3: BD Pharmingen), PE-labeled anti-CD133 (13A4: e-Bioscience), PE-labeled anti-CD 29 (Ha2/5: BD Pharmingen), APC-labeled anti-CD90 (30-H12: BioLegend), and APC-labeled CD106 (429: BioLegend), respectively.[…]FITC-labeled rat anti-mouse IgG2a, κ for Sca-1(e-Bioscience) was used.”

*The authors show that almost all spleen cells were donor-derived in matched BMT, whereas the majority of them were host-derived cells after mismatched BMT using immunofluorescence (Figure 2). However, this result is not enough to establish this point. The authors should show the flow cytometry results, using GFP or anti Ly-9.1 antibody. Since anti Ly-9.1 antibody react with lymphocyte of Blab/c mice but not B10.D2, this antibody is useful for distinguishing host cells from donor cells.*

Thank you for the comments and suggestions. We performed FACS analysis on spleen cells as suggested. We added a new graph for FACS analysis in the new Figure 2. The frequency of donor derived GFP cells in spleen is less in allogeneic BMT compared with syngeneic BMT. Figure 2 was replaced with the new data.

We rewrote the Results as follows:

“Interestingly, almost all spleen cells were donor-derived in matched WBMT, whereas the remaining host-derived cells after mismatched WBMT were higher in mismatched WBMT. (Figure 2).”

The authors show a significant increase of recipient derived CD4^+^Th17 PBMC cells (Figure 4, Left). If this result is true, all of CD4^+^

*T cells produce high levels of IL-17. Furthermore, almost all of other cells also produce high levels of IL-17. Does it increase serum IL-17 levels in this mouse?*

We conducted two independent experiments for IL-17 ELISA using stored serum samples from the same mice of the original experiments shown in Figure 5. As shown in a new Figure 5, serum IL-17 concentration significantly increased in the serum from the adoptive transferred recipients group compared with the control group.

We revised the description in the Results section, figure legend of the new Figure 5. We also inserted the new Figure 5 in the original Figure 5 as follows:

“We conducted 1L-17 ELISA using the serum samples from the same mice of the experiments shown in Figure 4. IL-17 concentration significantly increased in the serum from the adoptive transferred recipients group compared with the control group (Figure 5).”

Figure legend for Figure 5 was modified.

“(D) 1L-17 concentration in the serum from the same mice of the experiments shown in Figure 4 (n = 4 each). Data from two independent experiments (A-D).”

Methods section was modified as follows:

“Serum Cytokine Analysis

Serum was collected from mice using retro-orbital beads (Chen, et al., 2009). The concentration of IL-6 and IL-17 in the supernatants was determined by ELISA using a specific kit, according to the manufacturer’s instructions (BD Biosciences for IL-6, and R&D systems for IL-17). The experiments were performed in triplicate.”

Figure 1*unlike what is stated in the text, and enumerated in Figure 1, there do appear to be fibrotic changes in the lacrimal gland (albeit minimal), and in the salivary gland and perhaps in the intestine when BALB/c mice received B10 HSCs.*

The areas of minimal fibrotic changes in the lacrimal gland, salivary gland, and intestine when BALB/c mice received B10D2 HSCs are physiological, but not pathological changes and within normal limits. Collagen fibers (stained blue) physiologically support organ structure and large ducts in various tissues.

We added the description in the Results section as follows:

“The recipients of mismatched HSCs combined with syngeneic BMSCs were indistinguishable from syngeneic HSC and syngeneic BMSC recipients. Physicological connective tissues are also stained in blue in the lacrimal gland, salivary gland, and intestine when BALB/c mice received mismatched HSCs and syngeneic BMSCsFigure 1 and Figure 1—figure supplement 3).”

There also appears to be fibrosis in the lacrimal gland, salivary gland lungs, perhaps in liver and intestine in the mice receiving bone marrow depleted of PαS cells.

The fibrotic areas in the lacrimal gland, salivary gland, lungs, liver and intestine in the mice receiving bone marrow depleted of PαS cells are within normal range. Minimal Mallory staining areas are physiological changes and necessary to support the structure of duct and intestinal walls. We explained these issues in the Results and figure legends and added a new figure of target organs from normal mouse as Figure 1—figure supplement 4.

“Minimal Mallory staining areas are physiological changes and are necessary to support the structure of ducts and intestinal walls (Figure 1—figure supplement 4).”

In figure legend of Figure 1 and Figure 1—figure supplement 3 as follows:

“(B, i-v) Excessive fibrosis (deep blue, and＊) in various organs was observed in mismatched BMSC transplanted mice after 3 weeks.”

“D; duct. Minimal Mallory staining areas (▲) in (iii) and (V) are physiological changes and necessary to support the structure of ducts and intestinal walls.”

*Fibrotic lesions in all five groups were only analyzed in the lacrimal glands.*

We added the analyses for fibrotic lesions in all five groups in salivary gland, lung, skin, liver, and intestine. The figure was added as a new Figure 1—figure supplement 3.

We added the legend for Figure 1—figure supplement 3 as follows:

“(A) Mallory staining of salivary gland, skin, lung, liver and intestine tissue sections of BALB/c mice that received mismatched (ii) or syngeneic (iii) MSC transplantation and negative (i) or positive control (iv) at 3 weeks after transplantation.[…]Error bars indicate s.d. #p< 0.05, ∗p< 0.01, ∗∗p< 0.001. Scale bar, 100 μm.”

*It is stated that mismatched BMSC-induced fibrosis was dose dependent, but it only appears to be statistically significant when a dose that is 3.2x higher than the starting dose is used. The authors report using 10,000 BMSCs routinely, and this falls between the 8,000 and 16,000 cells used in Figure 1.*

We added data from 500, 2000, and 8000 syngeneic BMSCs transplantation in Figure 1. The differences between mismatched and syngeneic BMSCs transplantation were statistically significant between the two groups at each dose.

*In Figure 2, colocalization of GFP and HSP47 cannot be well seen in the lacrimal gland and in the intestine.*

We added arrows to show the GFP^+^ HSP47^+^ double positive fibroblasts in yellow (co expressions of GFP in green and HSP47 in red). The double positive cells exist in the lacrimal gland and intestine. However, the number of double positive cells for GFP and HSP47 in lacrimal gland was less than skin and intestine.

We revised the figure legend of Figure 2 as follows:

“Arrows indicate colocalized cells in yellow (GFP labeled BMSCs expressed HSP47) in lacrimal gland and intestine.”

*In 2C where the cells are syngeneic, very few GFP^+^ cells are seen in the lacrimal gland and none are seen in the skin and intestine, which makes one ask why the cells seem to survive in the mismatched experiments.*

We appreciate the editors’ and reviewers’ comments. We speculate that once mismatched BMSCs migrate into target organs via certain homing signals, mismatched BMSCs encounter host T cells and proliferate under the pathological microenvironment. This phenomenon may not be present in syngeneic BMSCs transplantation where an allogeneic response does not occur. As for HSP47 expression, it is clearly seen in activated fibroblasts, but only very faint expression is detected in quiescent fibroblasts in the syngeneic BMSC-transplanted recipients’ target organs.

We added the interpretation for the reviewers’ comments in the Discussion section as follows:

“Our findings suggest that once mismatched BMSCs migrate into the target organs via certain homing signals, mismatched BMSCs encounter T cells and proliferate and activate under the pathological microenvironment (Figure 2). T[…]As for HSP47 expression, it is clearly seen in activated fibroblasts, but only very faint expression is detected in quiescent fibroblasts in the syngeneic BMSC-transplanted recipients’ target organs.”

*Colocalization of GFP and HSP47 cannot be well seen in the in vitro culture of lacrimal fibroblasts.*

We modified the original colocalized figures to split images that better show both GFP in green and HSP47 in red in cultured fibroblasts derived from mismatched BMSC-transplanted lacrimal gland. We added the figures as new Figure 2.

*Figure 3—figure supplement 1 is more rightly a supplement for Figure 2.*

We moved Figure 3—figure supplement 1 to Figure 2—figure supplement 1 as suggested.

*In Figure 2, what is missing is the enumeration of spleen cells in the syngeneic and mismatched WBMT mice in order to say that the cells were recipient in nature.*

We added enumeration data of spleen cells from the syngeneic and mismatched WBMT recipients. A new graph was added as Figure 2 which replaces the figures of cytospin procedure from the same samples of spleen cells.

We revised the Results as follows:

“Interestingly, almost all spleen cells were donor-derived in matched WBMT, whereas the remaining host-derived cells after mismatched WBMT were higher in mismatched WBMT. (Figure 2)”.

*In Figure 4, left panel, no GFP staining is visible, and IL-6 staining is very faint.* The GFP expressing-cells from BMSCs in green colocalize with IL-6-producing cells in red. The merged cells in yellow are shown in the left panel of Figure 4. We added arrows in the figure and explained that BMSCs are IL-6 producing cells in the figure and figure legends.

We revised the figure legend of Figure 4, left as follows:

“Th 17 cells (pink in A) and IL-6 producing cells (yellow in left panel of Figure 4, arrows) were observed in the lacrimal gland of mismatched BMSC transplanted mice 8 weeks after transplantation. Yellow cells in (B) are due to co-localization of IL-6 (red) and donor BMSCs (GFP), resulting in yellow.”

In addition, we added split images from both figures as Figure 4—figure supplement 1.

A split image of Figure 4 is shown in Figure 4—figure supplement 1.

“Figure 4—figure supplement 1. Split images of IL-6 immunostaining and GFP+ donor MSC/ HSC into BALB/c Mismatched MSC-transplanted recipient lacrimal gland (left) and mismatched HSC-transplanted recipient lacrimal gland (right) from Figure 4. D, Duct.”

*Figure 5 shows only the lacrimal gland (other organs are listed in the text).*

Thank you for your suggestions. We have shown the histologic findings of other target organs as a new Figure 5—figure supplement 1 in the text, figure legend, and figure as shown below.

We added legend of Figure 5—figure supplement 1.

“Figure 5—figure supplement 1. (A) Mallory staining of salivary gland, skin, lung, liver and intestine tissue sections of BALB/c mice that received adoptive transfer and control. Excessive fibrotic areas are shown in deep blue (∗).”

Figure 5 shows only the lacrimal gland (other organs are listed in the text). Unlike what is stated in the text (that CD4^+^/Th17^+^ cells were "markedly" elevated), Figure 5 shows that the increase in CD4^+^/Th17^+^ cells in the adoptively transferred recipients is very limited (0.2% compared to 51.2% shown in Figure 4 for mismatched BMSCs placed into BALB/c mice), yet the fibrosis after the adoptive transfer is marked. The authors need to provide an interpretation of this result.

Adoptively transferred T cells have already acquired the auto-reactive nature to activate recipient-derived MSCs as shown in the in vitro analysis in Figure 6 in the revised manuscript. We interpret this phenomenon that adoptive transferred T cells react directly with recipient BMSCs, leading to fibrosis in the recipient. The lack of proliferation due to initial activation explains this difference between adoptively transferred mice and the original mismatched-MSC transplanted mice.

We added this interpretation of this result in the Discussion section as follows:

“Adoptive transferred T cells react recipient BMSCs leading to recipient’ fibrosis (Figure 5) probably because adoptively transferred T cells has already acquired the autoreactive nature to activate the recipient derived MSCs as shown in vitro analysis from Figure 6 and Figure 6.”

In addition, we have already discussed in the Discussion section as follows:

“Although the auto-antigen responsible for the autoimmune type reaction still needs to be identified, a common antigen specifically expressed in BMSCs of B10.D2 and BALB/c with different isoforms or SNIPs, is one of the most probable candidates. Since naïve T-cells responded to PαS-BMSCs in vitro at low levels, we speculate that auto-reactive T cells that are only present in basal levels under normal conditions (Sakaguchi, et al., 2008) can recognize minor differences between B10.D2 and BALB/c BMSCs.”

*In Figure 6, some important statistical comparisons are missing, and are needed to support the statement that IL-6 secretion was stimulated by BMSCs from either strain of mice.*

We appreciate the reviewers’ comments. We added some important statistical comparisons into the new Figure 6. We move Figure 6 to Figure 6.

In the Results section the statement that IL-6 secretion was stimulated by BMSCs from either strain of mice was explained as follows:

“Interestingly, T cells also responded to matched (BALB/c) BMSCs (Figure 6), suggesting that recipient T cells acquired the auto-reactive nature via antigen spreading (Shlomchik, et al., 2007). In addition, either strain-derived BMSCs induced slight IL-6 secretion when reacted with naïve T cells from wild type mice (Figure 6). Both PαS-BMSCs and T cells from mismatched BMSC-transplanted recipients produce IL-6 (Figure 6).”

*The data in Figure 6 should be reorganized to match the order of presentation in the text.*

We appreciate the reviewers’ detailed suggestions. We corrected the data in Figure 6 to match the order of presentation in the Results section as follows:

“T cells derived from mismatched BMSC-transplanted recipients proliferated and produced IL-6 in response to PαS-BMSCs but not DCs (Figure 6).[…]Both PαS-BMSCs and T cells from mismatched BMSC-transplanted recipients produce IL-6 (Figure 6).

Since the reaction was significantly suppressed by anti-MHC-class II antibody (Figure 6), and CD4^+^ T cells were predominant over CD8^+^ T cells (Figure 6), it was suggested that PαS-BMSCs themselves act as antigen presenting cells via MHC-class II.”